# The *Mycobacterium tuberculosis* complex pangenome is small and shaped by sub-lineage-specific regions of difference

Mahboobeh Behruznia[1,2], Maximillian Marin[3], Daniel J Whiley[1,4], Maha Reda Farhat[3,5], Jonathan C Thomas[1], Maria Rosa Domingo-Sananes[1], Conor J Meehan[1,6]*

[1]Department of Biosciences, Nottingham Trent University, Nottingham, United Kingdom; [2]Institute of Microbiology and Infection, College of Medical and Dental Sciences, University of Birmingham, Birmingham, United Kingdom; [3]Department of Biomedical Informatics, Harvard Medical School, Boston, United States; [4]Medical Technologies Innovation Facility, Nottingham Trent University, Nottingham, United Kingdom; [5]Pulmonary and Critical Care Medicine, Massachusetts General Hospital, Boston, United States; [6]Unit of Mycobacteriology, Institute of Tropical Medicine, Antwerp, Belgium

*For correspondence:
conor.meehan@ntu.ac.uk

Competing interest: The authors declare that no competing interests exist.

## eLife Assessment

This **useful** study analyzed 335 *Mycobacterium tuberculosis* Complex genomes and found that MTBC has a closed pangenome with few accessory genes. The research provides **solid** evidence for gene presence-absence patterns which support the appending conclusions however, the main criticism regarding the dominance of genome reduction remains.

**Abstract** The *Mycobacterium tuberculosis* complex (MTBC) is a group of bacteria causing tuberculosis (TB) in humans and animals. Understanding MTBC genetic diversity is crucial for insights into its adaptation and traits related to survival, virulence, and antibiotic resistance. While it is known that within-MTBC diversity is characterised by large deletions found only in certain lineages (regions of difference [RDs]), a comprehensive pangenomic analysis incorporating both coding and non-coding regions remains unexplored. We utilised a curated dataset representing various MTBC genomes, including under-represented lineages, to quantify the full diversity of the MTBC pangenome. The MTBC was found to have a small, closed pangenome with distinct genomic features and RDs both between lineages (as previously known) and between sub-lineages. The accessory genome was identified to be a product of genome reduction, showing both divergent and convergent deletions. This variation has implications for traits like virulence, drug resistance, and metabolism. The study provides a comprehensive understanding of the MTBC pangenome, highlighting the importance of genome reduction in its evolution, and underlines the significance of genomic variations in determining the pathogenic traits of different MTBC lineages.

## Introduction

The *Mycobacterium tuberculosis* complex (MTBC) is the term used for *M. tuberculosis* and all its variants, many of which used to be separate species (*Riojas et al., 2018*). These bacteria are the primary cause of tuberculosis (TB) in humans and other animals. Worldwide, TB is the 13th leading cause of

death and is once again the top infectious killer following global efforts to combat the COVID-19 pandemic (*World Health Organization WHO, 2024*). The MTBC comprises both human-associated lineages (L1 to L10) and animal-associated lineages (La1, La2, La3, *M. tuberculosis* var *microti* and *M. tuberculosis* var *pinnipedii*) (*Coscolla et al., 2021*; *Zwyer et al., 2021*; *Guyeux et al., 2024*). The human-adapted MTBC exhibits a strong phylogeographical population structure, with some lineages occurring globally and others showing a strong geographical restriction (*Gagneux, 2018*). For example, L2 and L4 are widespread globally, with L2 dominating in Southeast Asia and L4 in Europe (*Napier et al., 2020*). L1 and L3 occur mainly in regions around the Indian Ocean (*Menardo et al., 2021*), L5 and L6 are highly restricted to West Africa (*de Jong et al., 2010*) and L7-L10 are exclusively found in Central and East Africa (*Blouin et al., 2012*; *Ngabonziza et al., 2020*; *Coscolla et al., 2021*; *Guyeux et al., 2024*). Animal-associated lineages refer to lineages of the MTBC that primarily infect and cause disease in animals but can also infect humans in close contact with infected animals (*Brites et al., 2018*). Little is known about the geographic distribution and host range of animal-associated lineages (*Zwyer et al., 2021*).

Various factors such as human migration, local adaptation to host population, and host immune response are thought to have contributed to the current phylogeographical distribution of human-adapted MTBC members (*Darwin and Stanley, 2022*; *Hershberg et al., 2008*). Additionally, different lineages of MTBC are known to differ in virulence, metabolism and transmission potential, which can affect the spread of certain lineages and sub-lineages (*Coscolla, 2017*; *Gagneux et al., 2006*). Understanding the genetic diversity within MTBC is essential for gaining insights into how these bacteria adapt to different host populations and environments, and identify traits that may be important for survival, virulence, and antibiotic resistance.

MTBC is recognised for its clonal population structure and little genetic diversity between different strains, caused by the absence of horizontal gene transfer and plasmids (*Godfroid et al., 2018*; *Chiner-Oms et al., 2022*). Single nucleotide polymorphisms (SNPs) and large sequence polymorphisms (LSPs) represent the source of MTBC genetic diversity (*Gagneux, 2018*). Within the MTBC, LSPs which are deletions in genomes relative to the reference genome H37Rv are referred to as Regions of Difference (RDs; *Bespiatykh et al., 2021*). These RDs are known to differentiate MTBC lineages from each other (*Gagneux, 2018*), but within-lineage diversity is thought to be mostly composed of SNPs (*Napier et al., 2020*). While some RDs have been described to be differentially present within some lineages (*Sanoussi et al., 2021*; *Bespiatykh et al., 2021*), these are thought to be rare and remain unexplored. RDs can surpass 10 kb in length and encompass groups of genes with diverse functions, which can be linked to factors like virulence and metabolic diversity (*Bespiatykh et al., 2021*; *Bottai et al., 2020*). Additionally, the *IS6110* element has a notable impact on the genetic diversity of the MTBC. Transposition of *IS6110* into various coding or regulatory regions, and genomic deletions through homologous recombination between proximal copies, can also translate into significant genotypic variation at strain level (*Alonso et al., 2013*; *Antoine et al., 2021*). These mechanisms can all contribute to a varying repertoire of genetic features, much of which remains underexplored.

The pangenome represents the set of all genes present in a species. It comprises both the core genome, which consists of genes shared by all members of the species, and the accessory genome, which includes genes present in some but not all members. The accessory genome is often split into shell (present in most but not all genomes) and cloud (present in only a few genomes). Pangenomes are further classified as either open (each new genome adds a significant number of new genes to the population's accessory genome) or closed (newly analysed genomes do not significantly increase the accessory genome size; *Brockhurst et al., 2019*).

Previous research has focused on pangenomic investigations of particular MTBC lineages (*Baena et al., 2023*; *Ceres et al., 2022*; *Sanoussi et al., 2021*) and a recent study investigated the pangenome of most MTBC lineages using protein sequences (*Silva-Pereira et al., 2024*). However, there is a gap in comprehensive pangenomic analyses encompassing both coding and non-coding regions. Despite the name, prokaryotic pangenome analyses tend to not focus on full genome comparisons, but rather on protein-coding sequences. This focus neglects non-coding RNAs and pseudogenes as well as essential elements in intergenic regions, such as promoters, terminators, and regulatory binding sites, despite their demonstrated role in selection and phenotypic impact (*Tonkin-Hill et al., 2023*). Moreover, the inference drawn from pangenomic analyses is often influenced by short-read datasets. It has been shown that using short-read datasets can artificially inflate the size of the pangenome

estimates, leading to incorrect assumptions about evolutionary scenarios (*Marin, 2024*). Here, we used a diverse collection of closed MTBC genomes including human and animal-associated lineages to develop our understanding of MTBC pangenome and evolution incorporating both coding and non-coding regions. The objectives of this study are: (i) to establish the true size of the MTBC pangenome using a highly curated dataset, (ii) to assess specific biological functions that are prevalent in the core and accessory genome, and (iii) to assess gene content variation at both lineage and sub-lineage levels. By focusing on sub-lineages, we aim to provide a detailed characterisation of genetic diversity and structural differences that influence MTBC evolution and functional traits long term. We find that the MTBC has a small, closed pangenome and genome reduction via RDs is the driving evolutionary force that shapes the genomic repertoire of the species. We identified distinct genomic features among lineages and sub-lineages, including sub-lineage specific RDs, that may account for variations in virulence, metabolism, and antibiotic resistance.

## Results

### Overview of the genomic dataset

We combined 339 whole-genome sequences covering all known MTBC lineages except L10 for which no closed genome is available (*Guyeux et al., 2024*). This included 328 genomes published previously, as well as 11 genomes sequenced here from sub-lineages with few or no representatives in the public dataset to better cover the known phylogeny of MTBC (*Supplementary file 1*). These include the first closed genomes for L5.2, L9, La2 (formerly *M. caprae*) and *M. pinnipedii.* All genomes were sequenced to a depth greater than 30 x coverage, as required for >99% sequence recovery (*Sanderson et al., 2024*).

Except for Antarctica, our collection includes genomes from every continent, and different host species including humans, cattle, voles and BCG vaccine strains (*Supplementary file 2-A*). Africa has the highest diversity of MTBC genomes of all continents, which is consistent with the fact that the continent is known to have the most diverse MTBC strains (*Supplementary file 2-B* ; *Gagneux, 2018*). Most L1 genomes have been collected from East Africa, South and Southeast Asia, where this lineage is geographically common. L3 genomes were mostly isolated from India, while L2 and L4 genomes have been collected from different continents, representing the global distribution of these lineages. L5 and L6 genomes were exclusively found in West Africa, where they contribute significantly to the overall burden of TB in this region (*Coscolla et al., 2021*).

Across the total collection, genomes contained on average 3,950 protein-coding genes (CDS; range 3579–4018), of which 76 were predicted to be pseudogenes (range 64–93; *Supplementary file 1*). No significant association was found between the sequencing platform and pangenome size (Kruskal-Wallis test: p-value <0.001). Pairwise comparisons also found no significant differences in pangenome size (Dunn's test: p-value <0.001). However, a less stringent threshold (p-value <0.05) indicated a difference in pangenome size between Nanopore/Illumina hybrid assemblies and PacBio SMRT (no hybrid assembly), with Nanopore hybrid assemblies showing lower accessory gene counts. No difference in pseudogene count was observed based on sequencing technology, suggesting this was not impacted either.

**Table 1.** Pangenome (Panaroo) size estimations with and without corrections and merged paralogs. Corrected estimates are derived from filtering out false accessory genes through merging of genes with strong hits to the same gene in H37Rv (≥90% identity and ≥75% gene length coverage).

| Method | Pangenome statistics | Raw estimates | Corrected estimates |
|---|---|---|---|
| PanarooMerged paralogs | Total Genes | 4118 | 4032 |
| | Core genes | 3638 | 3627 |
| | Accessory genes: | 480 | 394 |
| PanarooIncluding paralogs | Total Genes | 4427 | 4321 |
| | Core genes: | 3635 | 3627 |
| | Accessory genes: | 792 | 694 |

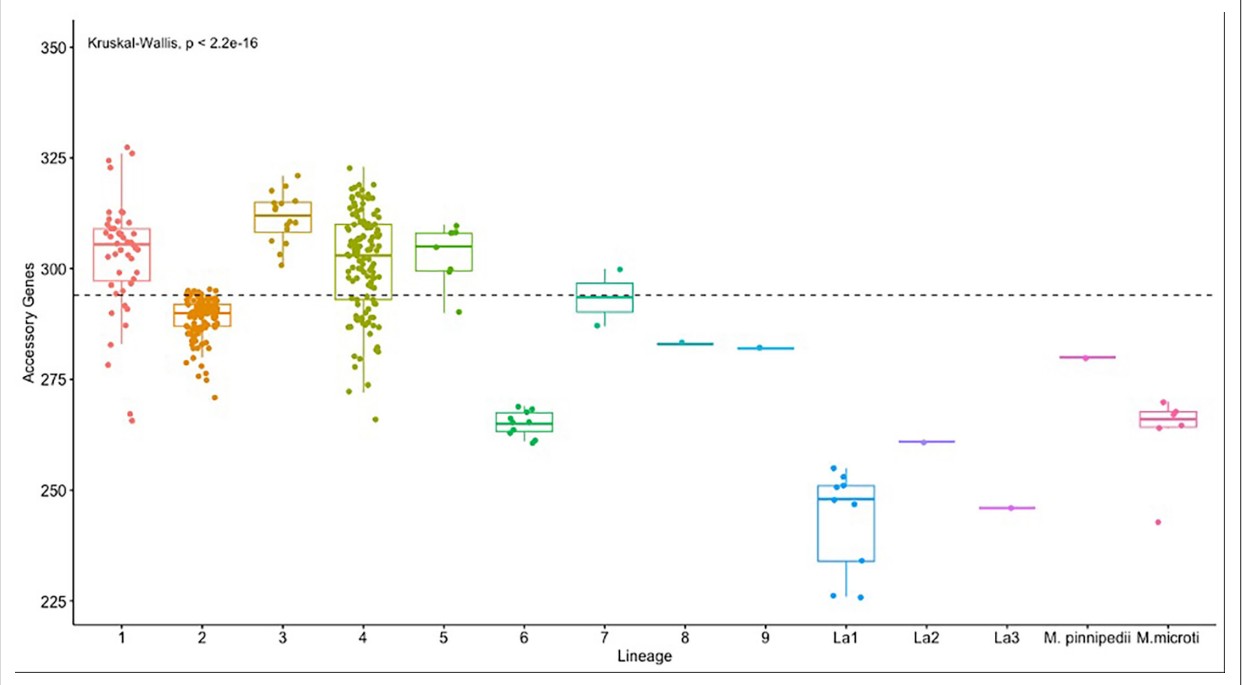

**Figure 1.** Boxplot showing the distribution of accessory genes within each lineage. Lineages with a small number of genomes were excluded from the statistical analysis. L6, La1, and *M. microti* have significantly smaller accessory genomes compared to other lineages.

## Pangenome characteristics and functional diversity

Panaroo estimated a protein-coding pangenome (with merged paralogs) consisting of 4,118 genes, including 3638 core genes (≥99% of genomes) and 480 accessory genes including 180 soft core (≥95% and<99% of genomes), 182 shell (≥15% and<95% of genomes), and 118 cloud (<15% of genomes) genes. After mapping all the genes in the pangenome to H37Rv to look for over-splitting of annotated genes, we identified 86 genes (2% of total genes) that were misclassified as accessory genes, including 9 out of 180 soft core, 12 out of 182 shell and 65 out of 118 cloud genes (*Table 1* and *Supplementary file 3*). This approach reduced the pangenome size from 4118–4032 genes and the accessory genome from 480 to 394 genes. Incorporating duplicated genes (i.e. not merging paralogs in Panaroo) increased the pangenome size to 4427 genes, including 3635 core and 792 accessory genes (195 soft core, 200 shell and 397 cloud). After combing genes based on mapping to H37Rv to reduce annotation splitting bias, the (including paralogs) pangenome size decreased to 4321 genes, including 3627 core genes and the accessory genome was reduced to 694 (187 soft core, 186 shell and 321 cloud) (*Table 1* and *Supplementary file 3*).

Heap's Law analysis resulted in an alpha of 2 for the merged paralog dataset and 1.548 for the unmerged dataset. This indicates a closed or near-closed pangenome based on both datasets. Genome fluidity was estimated to be 0.0085 for the merged paralog dataset and 0.0088 for the unmerged paralog dataset. This indicates any two randomly selected genomes only differ by 0.85–0.88% of gene content (*Supplementary file 4*).

All subsequent analyses based on Panaroo were undertaken using the merged paralog dataset, unless otherwise stated.

To examine genomic diversity among lineages, we investigated variations in the size of the accessory genome among different lineages. This analysis revealed significant differences in accessory genome size among different lineages (Kruskal-Wallis test, p=2.2e-16). Post-hoc Dunn tests with Bonferroni correction identified significant pairwise differences between lineages (*Supplementary file 5*). L6, La1, and *M. microti* have significantly smaller number of accessory genes per strain compared to others (*Figure 1*, *Supplementary file 5*); L2 has a significantly smaller accessory genome in comparison with L1, L3, and L4.

Functional annotation of core and accessory genes showed that genes of unknown function were the largest functional group in both, with over 20% (*Figure 2*). The core genome of MTBC encodes a high proportion of genes involved in lipid transport and metabolism (category I), transcription (category K) and cellular transport and secretion systems (category N), indicating the importance of these biological functions in defining the species. Functional breakdown of the accessory genome also revealed variations in genes involved in DNA replication, repair and mobile elements (category L) and lipid transport and metabolism (category I) across the species. The functional composition of core and accessory genes with the inclusion of paralogs is shown in *Figure 2—figure supplement 1*.

Pangraph revealed a similar distribution of core and accessory regions (*Table 2*). In total, 1338 regions were identified, with 1015 (75.8%) present in all genomes. The accessory regions were distributed as follows: 129 (9.6%) in the soft core, 140 (10.4%) in the shell and 54 (4%) in the cloud. Regions ranged in length from 250 bp (the set lower limit) to 54.6 kb (mean: 3.2 kb, median: 1 kb). Some regions (2.3% of total pangenome regions) were found to be mislabelled as absent in certain genomes, which were corrected using BLAST. This resulted in small corrections to the pangenome category distributions, primarily with an increase to the core region and reduction in the cloud accessory region count (*Table 2*), similar to the pangenome corrections above. Blocks that had these spurious hits discovered by BLASTn ranged in size from 254 bp to 5.4 kb (mean: 815 bp, median: 584 bp), suggesting this is an issue mainly impacting smaller Pangraph blocks.

After corrections, accessory blocks made up ~23% of the pangenome. These accessory blocks ranged in size from 250 bp to 9.8 kb (mean: 993 bp; median 629 bp). This suggests most accessory blocks each encompass only one or a few genes each, with most larger regions (i.e. all above 10 kb) being common to all the MTBC strains.

Heaps law based on the Pangraph block distribution resulted in an alpha of 2 and a genome fluidity of 0.041 (i.e. 4.1% uniqueness between any two genomes). This indicates a closed genome, similar to the Panaroo-based analyses, but with more unique blocks per genome when both coding and non-coding regions are considered compared to coding sequences alone (*Supplementary file 4*).

## Population structure analysis

The population structure was assessed using SNPs in the core genome (*Figure 3A*) and PCA plots were constructed based on presence-absence patterns in the accessory genome determined with Panaroo (includes coding genes only) and the accessory regions derived from Pangraph (includes both coding and non-coding genomic regions) (*Figure 3B–D*; *Supplementary file 6*). In the PCAs, genomes group together based on their lineage, indicating variations in accessory genome are mainly structured by lineage (*Figure 3B–D*). The first principal component (PC1) highlights the contrast between L2 from other members of MTBC. In all PCA plots, L2 formed a distinct group, defined primarily by the important variables RD105 and RD207 deletions. RD105 serves as a classic marker for all L2 genomes, and RD207 deletion is found in all L2.2 sub-lineages; the single L2.1 genome is separated from the L2.2 cluster due to the absence of RD207. The second principal component (PC2) demonstrates the difference between modern human-associated lineages (L2, L3, and L4) from other lineages. Important variables driving this separation are the TbD1 gene deletion in modern lineages, and gene deletions associated with RD11 (prophage phiRv2) in L6, L7, La1, and La3 genomes and some L1 sub-lineages. L1 genomes exhibit considerable genetic diversity primarily due to RD3 (related to prophage phiRv1) and RD11. The latter was identified in all L1.1 sub-lineages and certain L1.2 sub-lineages (L1.2.1, L1.2.2, and L1.2.2.2). The positioning of the single L8 genome (positive sign, *Figure 3B–D*) sits separate from all other lineages, likely due to being the most basal known lineage (*Ngabonziza et al., 2020*).

## The accessory genome of MTBC sub-lineages

To counteract the potentially misleading effect of mis-annotations of some genomes on accessory genome analyses, we looked for larger patterns of genomic differences in the MTBC, shared by multiple genomes. The accessory genome (Panaroo) and accessory region (Pangraph) patterns outlined above were filtered to look only at those genes and larger genomic regions present in all strains of one or more sub-lineages and completely absent in others (i.e. sub-lineage-specific genes and LSPs).

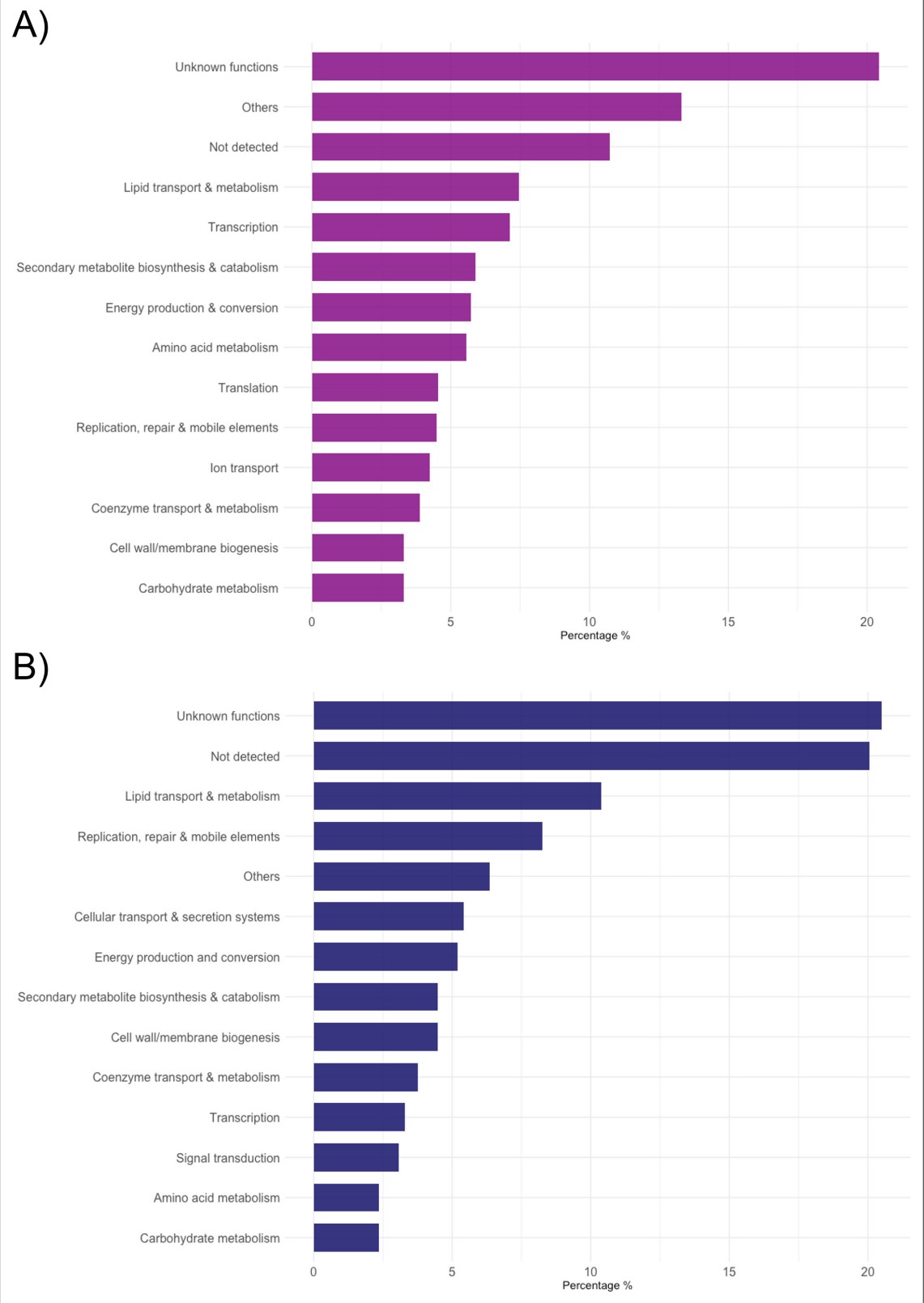

**Figure 2.** Analysis of the functional components in the core (**A**) and accessory (**B**) genome using EggNOG mapper and InterProScan. Core and accessory genes are derived from Panaroo with merged paralogs. For without merged paralogs, see *Figure 2—figure supplement 1*.

The online version of this article includes the following figure supplement(s) for figure 2:

**Figure supplement 1.** Analysis of the functional components in the core (**A**) and accessory (**B**) genome using EggNOG mapper and InterProScan.

**Table 2.** Pangenome graph (Pangraph) size estimations with and without corrections. Corrected estimates are derived from filtering out false labelling of absent regions using BLASTn.

| Pangenome partition | Raw estimates | Corrected estimates |
| --- | --- | --- |
| Total Regions | 1,338 | 1,338 |
| Core Regions | 1,015 | 1,040 |
| Soft core | 129 | 124 |
| Shell | 140 | 143 |
| Cloud | 54 | 31 |

Among the 394 filtered accessory genes, 111 were identified as specific to certain lineages and sub-lineages (*Supplementary files 7A and 8*). Four of these genes were absent from La3 but present in at least two copies in all other lineages: Rv3117 (cysA3) and Rv0815c (cysA2), involved in the formation of thiosulfate, and Rv3118 (sseC1) and Rv0814c (sseC2), involved in sulphur metabolism. Panaroo failed to identify partial gene deletions caused by RD9, RD10, RD13, RD105, RD207, and RD702. For example, RD702 is present (i.e. deleted) in all L6 and L9 genomes and covers a 641 bp region (*Figure 4*). RD702 affects less than a quarter of the *bglS* gene (470 bp from 2076 bp) and *mymT* gene (126 bp from 162 bp gene) in the 5' to 3' direction as well as a regulatory RNA (ncRv0186c) in the 3' to 5' direction. The Pangraph approach identified most partial gene deletion and non-coding regions of the DNA that were impacted by genomic deletion (relative to H37Rv). However, the graph approach failed to identify four structural variants, including a gene deletion corresponding to RD8 (Rv3618) in L6 and the animal-associated genomes.

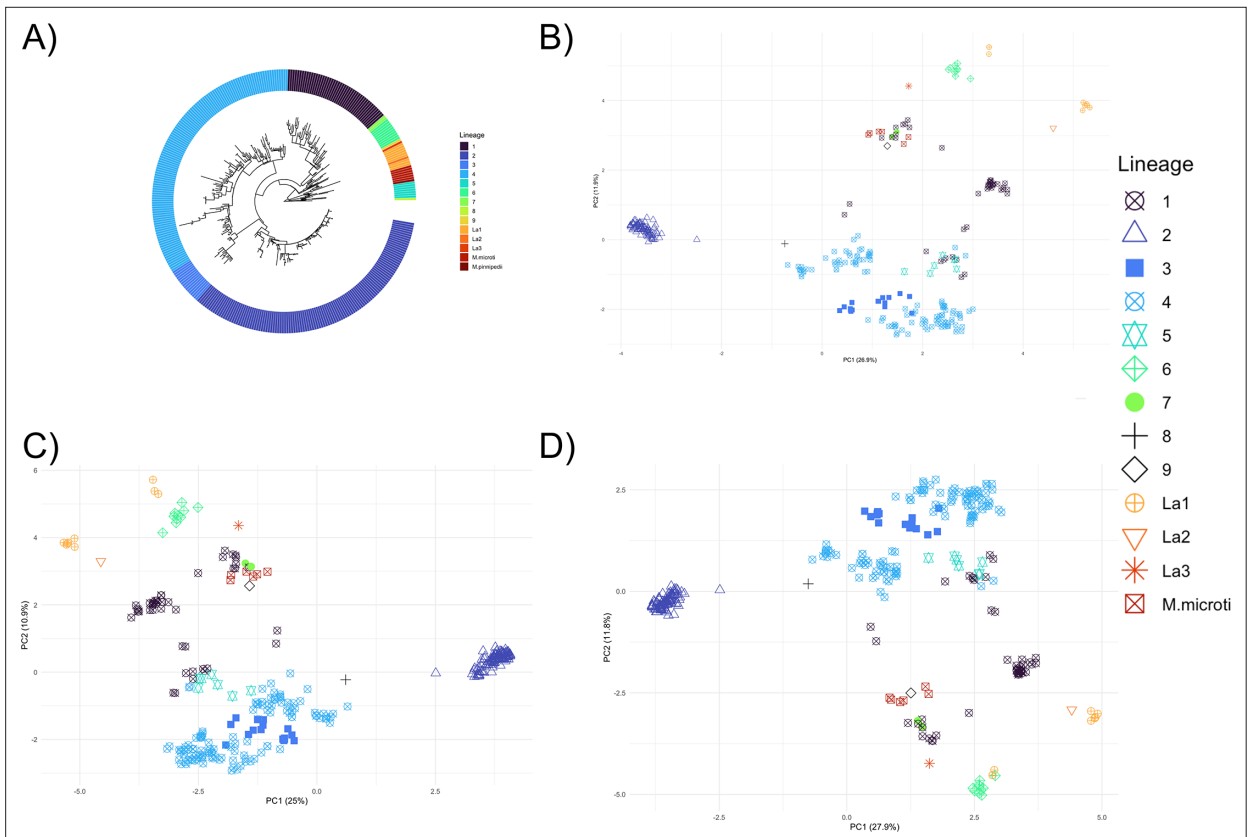

**Figure 3.** Population structure of the MTBC derived from phylogenetic and gene presence/absence approaches. (**A**) Phylogenetic tree based on MTBC core genome. PCA based on the accessory genome data from (**B**) Panaroo (merged paralogs), (**C**) Panaroo (no merged paralogs) and (**D**) accessory regions data from Pangraph.

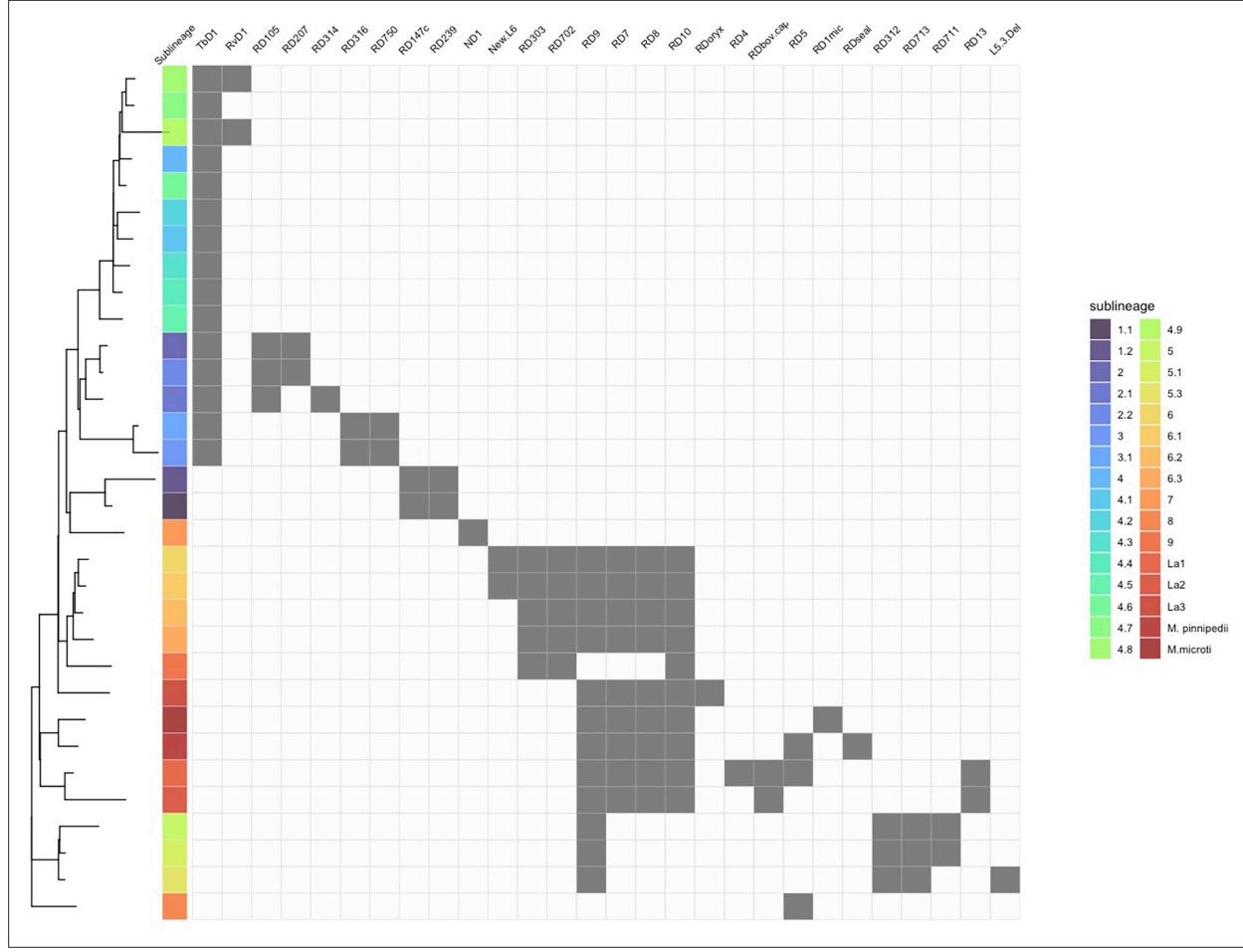

**Figure 4.** Sub-lineage specific regions of differences (RDs) identified using pangenome-based approaches. Sub-lineages are shown on the Y-axis coloured as per the legend. H37Rv sits within L4.9 at the top of the Y-axis. RDs (structural variants present in all members of one or more sub-lineages and absent in all members of one or more other sub-lineages) are listed on the X-axis, grouped by their pattern of presence/absence. Only RDs detected by both the Panaroo and Pangraph-based approaches are shown. Grey boxes indicate that the region is absent from that sub-lineage. Only RDs that are due to divergent evolution and found present or absent in 2 or more genomes are shown here.

Gene association analysis using coinfinder confirmed that most accessory genes (as determined by Panaroo) co-occur in pairs or groups in larger multi-gene RDs (as determined by Pangraph; *Supplementary file 7*). Thus, the small accessory genome is contained within an even smaller number of RDs. Coincident genomic blocks that were associated with lineages were RD8, RD9, RD10, and RD178, whereas those that were distributed across the phylogeny belonged to RD3 and RD11.

The filtered Pangraph results identified 82 lineage and sub-lineage-specific genomic regions ranging from 270 bp to 9.8 kb, encompassing a total of 116 genes and 4 non-coding RNA (*Supplementary file 8*). Many known lineage and sub-lineage-specific genomic regions were consistently identified using both methods (Panaroo and Pangraph) and found to be differentially present in two or more genomes, including TbD1, RD9, RD10, RD13, RD105, RD303, RD312, RD316, RD702, RD711, RD713, RD750, ND1, RDoryx, and N-RD25bov/cap (*Figure 4*).

Sub-lineage-specific accessory regions were classified as either RDs (deletions relative to H37Rv; present in most sub-lineages but absent in less than 50%) or LSPs (insertions/duplications; present in less than 50%). All sub-lineage-specific accessory regions were found to be RDs, i.e. present in H37Rv (L4.9 in *Figure 4*) and absent in only a minority of sub-lineages. Additional analysis of paralogous blocks, those present in all genomes but in multiple copies in some, did not reveal any new sub-lineage-specific accessory regions. Overall, these findings point towards deletion-driven evolution of the MTBC.

## Discussion

### The MTBC accessory genome is small and prone to inflation due to mis-annotation

The MTBC pangenome size has been reported to vary across studies. In a recent investigation conducted by *Marin, 2024*, the accessory genome ranged from 314 to 2951 genes as a function of the assembly quality, software and specific parameters used. Genome sequencing using short-read technology has some limitations that can result in incomplete assemblies, particularly in regions with repetitive sequences that are difficult to sequence (*Meehan et al., 2019*). About 10% of the coding sequences of the MTBC genome are composed of repetitive polymorphic sequences known as the PE (Proline-Glutamic acid) and PPE (Proline-Proline-Glutamic acid) families, implicated in host-pathogen interactions and virulence (*Cole et al., 1998*). Incomplete assemblies can impact the accurate detection of genes and may lead to artificial inflation of the estimated pangenome size (*Ceres et al., 2022*; *Marin, 2024*). We used a curated dataset to address these issues and estimate the true size of the MTBC pangenome. We showed that the MTBC has a compact accessory genome, totalling 10% of the total pangenome size (16% when including paralogs). This finding is consistent with recent research, which found a small accessory genome with a clonal genetic structure in the MTBC (*Baena et al., 2023*; *Ceres et al., 2022*; *Silva-Pereira et al., 2024*). Using a BLASTn analysis approach, we identified core genes that were designated as accessory and further reduced the accessory genome from 480 to 394 genes, a reduction of almost 20%. For example, two genes labelled as *lppA* and group_1645, both annotated as lipoproteins, were initially classified as accessory genes. However, these two genes exhibited 98% query coverage and 94% identity to the same gene in H37Rv, leading to their classification as a single core gene. Notably, 50% of the cloud genes (genes present in less than 15% of strains) were found to be core genes after this correction. These genes mapped to the reference genes using the BLAST approach and appeared to be fragmented genes caused by over-splitting in the PGAP annotation step.

The graph-based approach employed by Pangraph, encompassing all coding and non-coding regions, found a slightly larger accessory genomic content, with 22% of the total pangenome lying within accessory regions, after correction for spurious hits using BLASTn. This indicates that there is more variation to be uncovered when both coding and non-coding portions of the genome are assessed. However, correction of the Pangraph present/absent region patterns was required, showing caution and additional scrutiny is needed when assessing accessory region counts, especially for smaller blocks.

Identification of RDs was also likely impacted by the pangenome construction pipelines. Many partial RDs known to occur in specific lineages and sub-lineages were not identified by Panaroo. These results could be explained by the fact that Panaroo uses a relaxed alignment threshold along with neighbourhood gene information to cluster gene families (*Tonkin-Hill et al., 2020*). This can lead to the grouping of partially deleted genes with those that are complete. Pangraph appeared to overcome many of these issues as it uses the whole genome information as opposed to coding genes only. Despite these smaller limitations, both methods yielded consistent results indicating that the accessory genomic content (including coding and non-coding regions) of the MTBC is small, with much of the accessory genome contents shared between multiple strains (i.e. in the soft core and shell compartments). However, this does indicate that moving towards analysis of all genome regions, such as pangenome graphs, is required to identify the true accessory RD within a population, especially if that population has a near closed pangenome.

### The MTBC accessory genomic diversity is driven by reductive evolution

We employed a pangenomic approach to analyse the genetic diversity within the MTBC, aiming to understand its evolution and functional potential. In addition to inferring a gene-based pangenome with Panaroo, we also analysed variation in non-coding regions by inferring a pangenome graph with Pangraph. Examining non-coding regions is particularly important for species with limited genetic diversity, as they play a vital role in controlling gene expression, stress response and host adaptation (*Arnvig et al., 2014*). Our study demonstrated that, although both methods produced reliable outcomes, the whole-genome alignment approach is needed to capture all genetic variations, especially those involving large structural variants (>200 bp), as much variation appeared to be outside of protein coding genes. Moreover, this method identified non-coding regions of the genome that were

affected by genomic variation. Results from both methodologies revealed that the accessory genomic content of MTBC is a product of reductive evolution, that can be classified into divergent deletions arising from a single event (RDs) and convergent deletions, where the same deletion arises multiple times from independent events, with the former category being the more common. Coinfinder results also showed that coincident genes in the accessory genome primarily belonged to particular lineages.

However, a limitation of this analysis is that syntenic changes (those at the same places in different genomes) were not accounted for. Therefore, insertions or deletions which do not radically change the overall copy number but change the genome architecture could still occur and contribute to the overall evolutionary trajectory of the species. A more in-depth structural variation analysis accounting for such potential rearrangements is needed to assess the full impact of large insertions and deletions on the *M. tuberculosis* pangenome.

RDs, which are large deletions often encompassing multiple genes, can act as distinctive markers not only for lineages, as demonstrated previously, but also for sub-lineages. These warrant further investigation to understand the biological consequences of such deletions on traits like virulence, drug resistance, and metabolism, along with SNP only analyses. On the other hand, we observed several genomic deletions that appear to be due to convergent evolution and may indicate gene instability (*Brosch et al., 2002*). This includes RD3 and RD11 corresponding to the two prophages phiRv1 and phiRv2. One drawback of our study was that many of the lineages were poorly represented, especially in the L5, L6, L9, and all animal-associated lineages. The production of more closed genomes from the geographic regions where these are abundant (e.g. East and West Africa) will provide further information about the RD distributions within the MTBC and better understanding of the driving evolutionary forces.

All analysis undertaken here points towards genomic deletion as the main evolutionary factor creating the non-SNP-based variation within the MTBC. Reductive evolution is a driving factor of the other two *Mycobacterium* obligate pathogens, *M. ulcerans* and *M. leprae* (*Stinear et al., 2007*). The *M. tuberculosis* genome has also been suggested to be a product of genome reduction as its genome is much smaller than its sister species' (*Orgeur et al., 2024*). Our findings fit within this context, indicating that within-MTBC long-term evolution also tends to be reductive, with all discovered (sub-) lineage RDs being deletions relative to the complex as a whole. Much of this reduction appears to occur within the animal-associated lineages and the closely related human-associated lineages L5/L6/L9 (*Figure 4*). This is not to say duplication is not also an important factor in driving evolution within this group. It has previously been shown that duplication of genes implicated in drug resistance and virulence genes (e.g. PE/PPE genes) is common within the accessory genome, albeit in a lineage-independent manner (*Espinoza et al., 2025*). While accounting for duplicated genes did increase the accessory genome size here, especially the cloud genome (*Table 1*), this did not translate into significantly different patterns between sub-lineages. More closed genomes of closely related strains are required to look for duplication-driven evolution in shorter timescales, especially if this is associated with changes in virulence.

## The virulence potential of sub-lineages is differentially impacted by their accessory genomic content

The size of the accessory genome content varied significantly among different lineages. L2 genomes exhibited a high degree of clonality and a smaller accessory genome. L2 strains are characterised by a large number of *IS6110* copies, possibly due to the RD207, which affects the CRISPR-Cas type III-A system, responsible for adaptive immunity against mobile genetic elements (*Alonso et al., 2013*). While RD207 encompasses seven genes (Rv2814c-Rv2820c; *Bespiatykh et al., 2021*), our analysis revealed that only two genes, Rv2819c and Rv2820c, are specifically absent in L2 and can serve as a genetic marker for this lineage. Other RDs affecting all L2 genomes are RD3 (9.2 kb), and RD105 (3.4 kb). The latter is exclusively present in all L2 genomes and previous research proposed a potential role of RD105 in conferring drug resistance in L2.2 strains (*Qin et al., 2019*). RD207 (7.4 kb), caused by the recombination of two *IS6110* elements, is also present in all L2 genomes except for L2.1. L3 genomes exhibited significantly less genomic deletion in comparison with L2, and RDs affecting all L3 genomes were RD316 (1.2 kb) and RD750 (765 bp). The genes affected by these RDs are involved in lipid metabolism (category I) and secondary metabolite biosynthesis and transport respectively (category Q), which may contribute to metabolic variations in L3 strains. L4 exhibits substantial genetic

diversity primarily due to variations in the presence and absence of specific RDs at the sub-lineage level. For example, RD145 (4.7 kb) and RD182 (6.5 kb) were detected in L4.1.2.1 genomes isolated from Africa, Asia, Europe, and South America. However, none of the affected genes appears advantageous or essential for MTBC growth and survival, and this sub-lineage is successful worldwide (https://tbdr.lshtm.ac.uk/sra/browse). Other L4 sub-lineage-specific RDs detected in this study with no known advantage are RD115 in L4.3.4, RD727 in L4.6.1, RD219 in L4.8, RvD1 in L4.8 and L4.9. RD4 in La1, L5.3 Del in L5.3, and RD750 in L3, RD147c in L1.

Nonetheless, some genomic deletions can affect the genes which confer growth advantages in various lineages. Disruption of these accessory genes has been shown to provide a fitness advantage for the in vitro growth of H37Rv by analysis of saturated Himar 1 transposon libraries (*DeJesus et al., 2017*). For example, RD174 specific to L4.3.4 intersects with RD743 specific to L5, and both deletions affect two genes (Rv1995 and Rv1996) including *cmtR* transcriptional regulator. The latter regulates bacterial stress response to metal deficiency and is important during persistence in phagocyte (*Flores-Valdez et al., 2020*). While the absence of these genes may not be necessary for in vitro growth (*DeJesus et al., 2017*), the impact of this deletion may be different in in vivo conditions. The low prevalence of L4.3.4 strains in the TB Profiler database may suggest either the lower fitness of this sub-lineage or its presence in under-sampled geographic areas. RDs impacting the growth-advantage gene group were prevalent in L5 genomes (L5.3 Del, RD743, and RD711). For instance, RD711 affects the Rv1335 gene, which is annotated as a sulphur carrier protein involved in cysteine biosynthesis by PGAP. This may imply that the genome reduction observed in L5 has led to dependence on host factors for essential nutrients, consequently making these strains difficult to grow in a culture medium (*Sanoussi et al., 2017*).

L6 and animal-associated lineages accumulated a higher number of deletions including RD7, RD8, RD9, and RD10. Comparative analysis of the accessory genome size in L6, La1, and *M. microti* revealed significant differences compared to other lineages. The identified genomic regions in L6, L9, and animal-associated lineages impact genes associated with virulence, potentially explaining their reduced virulence in human hosts. RD7 contains the mammalian cell entry (*mce3*) operon implicated in the invasion of host cells, while RD9 includes genes related to biliverdin reduction, potentially aiding in protection against oxidative stress inside macrophages (*Szklarczyk et al., 2023*). RD10 impacts genes related to lipid storage crucial for MTBC transition to a dormant state (*Deb et al., 2009*). RD1BCG and RD1mic affect genes encoding the ESX-1 secretion system, crucial for virulence and phagosomal escape, with RD1BCG being deleted during BCG vaccine development (*Szklarczyk et al., 2023*). Overall, the findings of this study suggest that, despite having a compact accessory genome, it can account for certain biological distinctions among MTBC. While genetic drift has been proposed as a key driver of diversity in MTBC (*Hershberg et al., 2008*), recent research has argued that it does not fully explain the observed low genetic diversity in MTBC. Instead, purifying selection may be the dominant force, removing harmful mutations and maintaining genetic uniformity (*Stritt and Gagneux, 2023*).

## A need for more high-quality genomes and annotations

While this dataset represents high-quality MTBC genome sequences that cover most of its known diversity, it suffers from sampling bias as some lineages are over-represented whereas others have a few representatives. The over-representation of L2 and L4 in the dataset is primarily attributed to their association with Bioprojects analysing TB outbreaks or originating from countries with extensive sampling, highlighting a bias in sequencing efforts towards more prevalent or historically studied lineages. Many sub-Saharan African countries do not have a representative sample in our dataset, and this may explain the under-representation of L5 and L6 in the dataset. These lineages contribute significantly to the overall burden of TB in West Africa (*Coscolla et al., 2021*). Additionally, we found that certain RDs that were previously assumed to be specific to certain lineages were identified in other lineages (*Supplementary file 8*). The observed variation in RD distribution emphasises the need for an expanded genomic dataset to provide a more comprehensive understanding of their prevalence, specificity, and implications in the evolution of the MTBC. Future sequencing efforts should target high-burden TB countries, ensuring a comprehensive representation of the population structure to enhance our understanding of MTBC diversity and evolution. Closed genomes for closely related strains are also required to better understand short-term evolutionary patterns and the impact upon

virulence within circulating strains. Additionally, investigating human transcriptomic data could further improve our understanding of the MTBC host adaptation, geographic distribution and pathogenicity.

Alongside these genomes, highly accurate annotation of the genes within is also needed. The availability of the H37Rv genome, which has undergone many rounds of intense reannotation (*Modlin et al., 2021*), allowed for merging of genes that were over-split by PGAP. However, this meant only genes present in H37Rv could be re-merged where needed, potentially leaving other genes mis-annotated in the accessory genome. High-quality, and high-throughput annotation of other genomes will help rectify this in future expanded studies.

## Conclusion

In conclusion, this study reveals that the MTBC has a small, near-closed pangenome, predominantly driven by genome reduction. This observation underlines the importance of genomic deletions in the evolution of the MTBC. We found many distinct genomic features across the MTBC, including sublineage-specific RDs, which likely play a crucial role in variations in virulence, metabolism, and anti-biotic resistance. Finally, the study underscores the impact of non-coding regions and large structural variations on the MTBC's genetic diversity, offering a new perspective on MTBC pangenomics.

## Methods

### Summary of sequencing data and quality control

Previous studies have shown that closed genomes should be used for the best estimation of pange-nome size (*Marin, 2024*). From a combination of previously published research and publicly available data, 328 non-duplicated, complete (single contig) and closed (as listed on NCBI) genome assemblies were selected for having (i) long-read sequencing only (Nanopore or PacBio) or (ii) hybrid sequencing with Illumina across the MTBC lineages. Information related to the geographical location, source of isolation, assembly accessions and quality metrics can be found in *Supplementary file 1*. The quality of all assemblies was assessed using BUSCO v5.8.2 (*Manni et al., 2021*) and only assemblies with ≥95% completeness were used for further analysis. For parity, all assemblies were set to start on the *dnaA* gene using the fixstart command of Circlator v1.5.5 (*Hunt et al., 2015*). Complete circular assemblies were annotated using PGAP v6.4 with default options (*Tatusova et al., 2016*). This approach is recommended by previous analyses (*Marin, 2024*) as PGAP tends to annotate pseudogenes in cases of pseudogenisation and frameshift, rather than annotating them as new coding sequences. Visualisations of the PGAP annotation were made using the DNA Features Viewer Python library (*Zulkower and Rosser, 2020*). Lineage classification was performed using TB-profiler v6.5 with the TBDB v.72ef6fa database (*Phelan et al., 2019*) except for *M. tuberculosis* var *microti* (hereafter named *M. microti) and M. tuberculosis* var *pinnipedii* (hereafter named *M. pinnipedii*) which are not classified in this database and were assigned using spoligotype using SpolPred2 as implemented in TB-profiler (*Napier et al., 2023*).

### Nanopore sequencing and assembly

We sequenced lineages and sub-lineages that either had no sequencing data available in public data-bases or were under-represented in the collection. These isolates belonged to L5 (n=2, L5.1 and L5.2), L4.6 (n=3, L4.6.1, L4.6.1.1 and L4.6.1.2), L4.7 (n=3), L9 (n=1), La2 (*M. caprae*, n=1) and *M. pinnipedii* (n=1). A list of genomes sequenced in this study and their assembly information can be found in *Supplementary file 1*. Raw reads and assembled genomes were deposited in NCBI under the Bioproject PRJNA1085078.

DNA extraction suitable for long-read sequencing was performed using the Genomic-tip 100 /G (Qiagen Inc) as described previously (*Ngabonziza et al., 2020*). DNA concentration was quantified using a Qubit fluorometer and 1 X dsDNA High Sensitivity (HS) assay kit (Thermo Fisher Scientific). Nanopore sequencing was performed using the native barcoding kit 24 V14 (SQK-NBD114.24) according to the manufacturer's instruction on the MinION Mk1C platform with R10.4.1 flow cells. Basecalling was undertaken using Guppy v 6.5.7 with the super accurate (SUP) model. Sequence adaptors were trimmed using Porechop v0.2.4 with '--end_threshold 95' and '--middle_threshold 85' options (https://github.com/rrwick/Porechop; *Wick, 2018*). Sequences were assembled using flye v2.9.2 with default options (*Kolmogorov et al., 2019*) and iteratively polished using Racon v1.5.0 and

the 'consensus' subcommand of Medaka v1.8.0 (https://github.com/nanoporetech/medaka; *Oxford Nanopore Technologies, 2024*). Complete assemblies underwent Circlator, BUSCO, and PGAP steps as above.

## Pangenome analysis (protein coding genes)

Panaroo v 1.5.1 inferred the MTBC protein-coding pangenome (*Tonkin-Hill et al., 2020*) with and without the '--merge_paralogs' option to investigate gene duplication (paralogy). Additionally, '--clean-mode strict' and '--core_threshold 0.95' were applied to reduce potential contamination by removing genes found at low frequencies (present in fewer than 5% of genomes) and to define core alignment gene set for the subsequent phylogenetics, respectively. *Ceres et al., 2022* showed that genes could be over-split by annotation software, creating additional accessory genes, which has been confirmed elsewhere (*Marin, 2024*). To counteract this, we followed the approach recommended in *Ceres et al., 2022* by merging genes that map to the same gene in a reference genome. The reference pangenome created by Panaroo was compared to the H37Rv genome (standard reference genome for the MTBC) using BLASTn with a similarity e-value cutoff of $1\times10^{-10}$ (*Zhang et al., 2000*). The BLASTn result was then filtered to only include queries with ≥90% identity and ≥75% of the H37Rv gene length covered using in-house Python scripts (https://github.com/conmeehan/pathophy; copy archived at *Meehan, 2025*). A core genome alignment was constructed using the built-in MAFFT aligner included in Panaroo (*Katoh et al., 2002*). The core phylogeny was created using RAxML-NG v1.2.2 with a GTR substitution model (*Kozlov et al., 2019*). Tree visualisations were created using the Treeio package v1.20.0 (*Wang et al., 2020*).

## Pangenome graph construction

The above pangenome analysis only compares coding gene content and omits all non-coding genes and intra-gene genetic content. To examine the total pangenome, including both coding and non-coding regions, all genomes were aligned into a graph using Pangraph v0.7.2 (*Noll et al., 2023*). A minimum genome region block size of 250 bp was selected to look only at large polymorphisms. As above, an additional check was undertaken on blocks labelled as absent, to ensure they truly are not present in the given genome, and not just mislabelled as absent by the Pangraph pipeline. BLASTn was used to compare the consensus sequence for that region and the genome it is predicted to be absent from. Any BLASTn results with an e-value less than $1e^{-30}$ and additionally covering 50% of the region length and a 50% identical match were re-labelled as present. Genomic regions were classified as a core region if present in all genomes or an accessory region if present in some but not all genomes. The DNA sequences from genomic blocks present in at least one sub-lineage but completely absent in others were extracted to look for long-term evolution patterns in the pangenome.

Known RDs were identified using RDscan (*Bespiatykh et al., 2021*). Structural variants with Rv (H37Rv gene) loci overlapping previously annotated RDs were assigned the same RD names. Novel structural variants that did not align with any previously reported RDs were classified as new RDs if they were present in H37Rv, or as a new LSP if absent. Accessory regions were further classified as a deletion if present in over 50% of the sub-lineages or an insertion/duplication if present in less than 50% of sub-lineages. This approach provides a reference-free perspective for investigating gene content differences and labels relevant LSPs as reductive (deletions) or increasing (insertions/duplications).

## Functional genome analysis

The reference pangenome created by Panaroo was translated using transeq v5.0.0 (*Rice et al., 2000*) to obtain the protein sequences. COG functional categories were assigned using eggNOG Mapper v5.0.2 using default options (*Huerta-Cepas et al., 2017*). Genes that did not return a match within the eggNOG database were included in the 'not found' category. Functional domain annotation was performed with InterProScan v5.63–95 (*Jones et al., 2014*). For the gene association analysis, the accessory genome (genes present in less than 95% of genomes) was used as input for Coinfinder v1.2.1 (*Whelan et al., 2020*). The STRING database (https://string-db.org/) was used to infer interaction networks of genes. Predicted functional partners with a high confidence interaction score (≥0.7) by STRING v12.0 were considered in this study.

## Statistical analysis

Statistical analysis was conducted using R studio v2024.04.2+764 (*R Development Core Team, 2023*). Due to the use of multiple public genomes created using different sequencing technologies, we looked for significant associations between accessory genome size and sequencing technology. For comparative purposes and to ensure sufficient sample size per group, all types of Illumina sequencing (e.g. MiSeq, HiSeq, etc) were combined, resulting in comparisons between different long-read technologies, with or without Illumina hybrid sequencing. Any sequencing technology group with fewer than three genomes was excluded. The Shapiro-Wilk test was used to test for normality and then a Kruskal-Wallis test was used to look for associations between accessory genome count (the number of missing genes in a genome) and sequencing platform(s). A Dunn's test was then used to look for pairwise significant differences with a Bonferroni correction for multiple comparisons. The same process was undertaken on pseudogene counts, based on the GFF files output from PGAP.

Principal component analysis (PCA) based on gene presence and absence data from Panaroo-derived accessory genome contents and Pangraph-derived accessory region contents was performed using the prcomp package in R. The fviz_contrib function in Factoextra (v1.0.7) was used to identify the contribution of important variables from the PCA results (*Kassambara and Mundt, 2020*).

To assess the openness and pairwise uniqueness of the pangenome, micropan was used to estimate Heaps law, genome fluidity and rarefaction curves with 100 permutations (*Snipen and Liland, 2015*). An alpha of 1 or lower in Heap's law indicates an open genome; higher than 1 indicates a closed genome.

## Acknowledgements

We thank all the staff at the BCCM/ITM collection for the growth of strains for sequencing in this study. We thank Alasdair Hubbard for assistance with Nanopore sequencing. CJM, DJW and MB are supported by the Academy of Medical Sciences (AMS), the Wellcome Trust, the Government Department of Business, Energy and Industrial Strategy (BEIS), the British Heart Foundation and Diabetes UK and the Global Challenges Research Fund (GCRF) via a Springboard grant [SBF006\1090].

## Additional information

### Funding

| Funder | Grant reference number | Author |
| --- | --- | --- |
| Academy of Medical Sciences | SBF006\1090 | Mahboobeh Behruznia<br>Daniel J Whiley<br>Conor J Meehan |

The funders had no role in study design, data collection and interpretation, or the decision to submit the work for publication.

### Author contributions

Mahboobeh Behruznia, Conceptualization, Data curation, Formal analysis, Investigation, Methodology, Writing – original draft, Writing – review and editing; Maximillian Marin, Resources, Data curation, Formal analysis, Methodology, Writing – review and editing; Daniel J Whiley, Data curation, Formal analysis, Methodology, Writing – review and editing; Maha Reda Farhat, Data curation, Supervision, Methodology, Writing – review and editing; Jonathan C Thomas, Resources, Methodology, Writing – review and editing; Maria Rosa Domingo-Sananes, Formal analysis, Validation, Investigation, Visualization, Methodology, Writing – review and editing; Conor J Meehan, Conceptualization, Formal analysis, Supervision, Funding acquisition, Investigation, Methodology, Writing – original draft, Project administration, Writing – review and editing

### Author ORCIDs

Mahboobeh Behruznia https://orcid.org/0000-0002-3302-7272
Maximillian Marin https://orcid.org/0000-0002-9108-3328
Daniel J Whiley https://orcid.org/0000-0002-3972-5100

Jonathan C Thomas [ID] https://orcid.org/0000-0002-1599-9123
Maria Rosa Domingo-Sananes [ID] https://orcid.org/0000-0002-3339-8671
Conor J Meehan [ID] https://orcid.org/0000-0003-0724-8343

Reviewer #1 (Public review): https://doi.org/10.7554/eLife.97870.4.sa1
Author response https://doi.org/10.7554/eLife.97870.4.sa2

## Additional files

### Supplementary files

Supplementary file 1. An overview of the genome dataset, both publicly acquired data and those strains sequenced within this study including assembly accessions and online locations, where relevant. BUSCO completeness information for each genome is also provided, as is the CDS and pseudogene count as predicted by PGAP.

Supplementary file 2. The geography of the dataset. (A) sample collection distribution by country; (B) the number of genomes of each lineage included from each continent.

Supplementary file 3. A list of all gene groups which were combined to reduce over-splitting of pangenome due to annotation errors. Each gene group's name is listed alongside its original classification (Core, Sort-core, Cloud, Shell) and its new classification after merging with others after annotation correction. This is shown for both the pangenome with and without merged paralog setting enabled.

Supplementary file 4. Pangenome openness assessment using Heap's Law analysis, genome fluidity and rarefaction curve for (A) Panaroo with merged paralogs; (B) Panaroo with unmerged paralogs and (C) Pangraph blocks.

Supplementary file 5. Pairwise comparisons of accessory genome size between lineages. A * indicates a significant difference between these lineages. Comparisons on both merged and unmerged paralog datasets are shown.

Supplementary file 6. MTBC accessory genome distribution based on (A) Panaroo (merged paralogs) and (B) Pangraph. The MTBC phylogenetic tree is shown on the left beside a coloured bar indicating the sub-lineage of each tip genome. The accessory genes/regions are indicated by columns in the heatmap with a blue box if present in that strain's genome.

Supplementary file 7. Coinfinder gene association heatmap. Coinfinder reveals gene association patterns within the accessory genome of the MTBC. Accessory genes, listed on the X-axis, cluster into groups (shown here by varying colours of the blocks).

Supplementary file 8. A list of all RDs (known and new) along with the genes contained in each and the lineages the regions are absent from.

MDAR checklist

### Data availability

The accession codes for the 328 previously sequenced publicly available genomes are listed in *Supplementary file 1*. The 11 newly sequenced genomes are deposited in NCBI in the BioProject PRJNA1085078. Code for processing BLAST results can be found at https://github.com/conmeehan/pathophy, copy archived at *Meehan, 2025*.

The following dataset was generated:

| Author(s) | Year | Dataset title | Dataset URL | Database and Identifier |
|-----------|------|---------------|-------------|-------------------------|
| Behruznia M, Whiley D, Meehan CJ | 2024 | Nanopore sequencing of *Mycobacterium tuberculosis* complex genomes | https://www.ncbi.nlm.nih.gov/bioproject/PRJNA1085078/ | NCBI BioProject, PRJNA1085078 |

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
