## [Editor Report · eLife Assessment]

This **useful** study analyzed 335 *Mycobacterium tuberculosis* Complex genomes and found that MTBC has a closed pangenome with few accessory genes. The research provides **solid** evidence for gene presence-absence patterns which support the appending conclusions however, the main criticism regarding the dominance of genome reduction remains.

---

## [Referee Report · Reviewer #1 (Public review)]

Summary:

In this paper, Behruznia and colleagues use long-read sequencing data for 339 strains of the *Mycobacterium tuberculosis* complex to study genome evolution in this clonal bacterial pathogen. They use both a "classical" pangenome approach that looks at the presence and absence of genes, and a pangenome graph based on whole genomes in order to investigate structural variants in non-coding regions. The comparison of the two approaches is informative and shows that much is missed when focusing only on genes. The two main biological results of the study are that (1) the MTBC has a small pangenome with few accessory genes, and that (2) pangenome evolution is driven by genome reduction. The second result is still questionable because it relies on a method that disregards paralogs.

Strengths:

The authors put together the so-far largest data set of long-read assemblies representing most lineages of the *Mycobacterium tuberculosis* context, and covering a large geographic area. They sequenced and assembled genomes for strains of M. pinnipedi, L9, and La2, for which no high-quality assemblies were available previously. State-of-the-art methods are used to analyze gene presence-absence polymorphisms (Panaroo) and to construct a pangenome graph (PanGraph). Additional analysis steps are performed to address known problems with misannotated or misassembled genes.

Weaknesses:

The main criticism regarding the dominance of genome reduction remains after two rounds of revisions. A method that systematically excludes paralogs is hardly suitable to draw conclusions about the relative importance of insertions/duplications and deletions in a clonal organism, where any insertion/duplication will result in a paralog. I understand that a re-analysis of the data might not be practical, and the authors have added a few sentences in the discussion that touch on this problem. However, the statements regarding the dominance of genome reduction remain too assertive given this basic flaw.

Here are the more detailed argument from the previous review:

In a fully clonal organism, any insertion/duplication will be an insertion/duplication of an existing sequence and thus produce a paralog. If I'm correctly understanding your methods section, paralogs are systematically excluded in the pangraph analysis. Genomic blocks are summarized at the sublineage level as follows (l.184): "The DNA sequences from genomic blocks present in at least one sub-lineage but completely absent in others were extracted to look for long-term evolution patterns in the pangenome." I presume this is done using blastn, as in other steps of the analysis.

So a sublineage-specific copy of IS6110 would be excluded here, because IS6110 is present somewhere in the genome in all sublineages. However, the appropriate category of comparison, at least for the discussion of genome reduction, is orthology rather than homology: is the same, orthologous copy of IS6110, at the same position in the genome, present or absent in other sublineages? The same considerations apply to potential sublineage-specific duplicates of PE, PPE, and Esx genes. These gene families play important roles in host-pathogen interactions, so I'd argue that the neglect of paralogs is not a finicky detail, but could be of broader biological relevance.

---

## [Author Response]

The following is the authors’ response to the previous reviews

**Public Reviews:**

**Reviewer #1 (Public review):**
Summary:In this paper, Behruznia and colleagues use long-read sequencing data for 339 strains of the *Mycobacterium tuberculosis* complex to study genome evolution in this clonal bacterial pathogen. They use both a "classical" pangenome approach that looks at the presence and absence of genes, and a pangenome graph based on whole genomes in order to investigate structural variants in non-coding regions. The comparison of the two approaches is informative and shows that much is missed when focussing only on genes. The two main biological results of the study are that (1) the MTBC has a small pangenome with few accessory genes, and that (2) pangenome evolution is driven by genome reduction. In the revised article, the description of the data set and the methods is much improved, and the comparison of the two pangenome approaches is more consistent. I still think, however, that the discussion of genome reduction suffers from a basic flaw, namely the failure to distinguish clearly between orthologs and homologs/paralogs.Strengths:The authors put together the so-far largest data set of long-read assemblies representing most lineages of the *Mycobacterium tuberculosis* context, and covering a large geographic area. They sequenced and assembled genomes for strains of M. pinnipedi, L9, and La2, for which no high-quality assemblies were available previously. State-of-the-art methods are used to analyze gene presence-absence polymorphisms (Panaroo) and to construct a pangenome graph (PanGraph). Additional analysis steps are performed to address known problems with misannotated or misassembled genes.Weaknesses:The revised manuscript has gained much clarity and consistency. One previous criticism, however, has in my opinion not been properly addressed. I think the problem boils down to not clearly distinguishing between orthologs and paralogs/homologs. As this problem affects a main conclusion - the prevalence of deletions over insertions in the MTBC - it should be addressed, if not through additional analyses, then at least in the discussion.Insertions and deletions are now distinguished in the following way: "Accessory regions were further classified as a deletion if present in over 50% of the 192 sub-lineages or an insertion/duplication if present in less than 50% of sub-lineages." The outcome of this classification is suspicious: not a single accessory region was classified as an insertion/duplication. As a check of sanity, I'd expect at least some insertions of IS6110 to show up, which has produced lineage- or sublineage-specific insertions (Roychowdhury et al. 2015, Shitikov et al. 2019). Why, for example, wouldn't IS6110 insertions in the single L8 strain show up here?In a fully clonal organism, any insertion/duplication will be an insertion/duplication of an existing sequence, and thus produce a paralog. If I'm correctly understanding your methods section, paralogs are systematically excluded in the pangraph analysis. Genomic blocks are summarized at the sublineage levels as follows (l.184): "The DNA sequences from genomic blocks present in at least one sub-lineage but completely absent in others were extracted to look for long-term evolution patterns in the pangenome." I presume this is done using blastn, as in other steps of the analysis.So a sublineage-specific copy of IS6110 would be excluded here, because IS6110 is present somewhere in the genome in all sublineages. However, the appropriate category of comparison, at least for the discussion of genome reduction, is orthology rather than homology: is the same, orthologous copy of IS6110, at the same position in the genome, present or absent in other sublineages? The same considerations apply to potential sublineage-specific duplicates of PE, PPE, and Esx genes. These gene families play important roles in host-pathogen interactions, so I'd argue that the neglect of paralogs is not a finicky detail, but could be of broader biological relevance.
**Reviewer #2 (Public review):**
Summary:The authors attempted to investigate the pangenome of MTBC by using a selection of state-of-the-art bioinformatic tools to analyse 324 complete and 11 new genomes representing all known lineages and sublineages. The aim of their work was to describe the total diversity of the MTBC and to investigate the driving evolutionary force. By using long read and hybrid approaches for genome assembly, an important attempt was made to understand why the MTBC pangenome size was reported to vary in size by previous reports. This study provides strong evidence that the MTBC pangenome is closed and that genome reduction is the main driver of this species evolution.Strengths:A stand-out feature of this work is the inclusion of non-coding regions as opposed to only coding regions which was a focus of previous papers and analyses which investigated the MTBC pangenome. A unique feature of this work is that it highlights sublineage-specific regions of difference (RDs) that was previously unknown. Another major strength is the utilisation of long-read whole genomes sequences, in combination with short-read sequences when available. It is known that using only short reads for genome assembly has several pitfalls. The parallel approach of utilizing both Panaroo and Pangraph for pangenomic reconstruction illuminated limitations of both tools while highlighting genomic features identified by both. This is important for any future work and perhaps alludes to the need for more MTBC-specific tools to be developed. Lastly, ample statistical support in the form of Heaps law and genome fluidity calculations for each pangenome to demonstrate that they are indeed closed.Weaknesses:There are no major weaknesses in the revised version of this manuscript.
**Recommendations for the authors:**

**Reviewer #1 (Recommendations for the authors):**
l. 27: "lineage-specific and -independent deletions": it is still not clear to me what a lineage-independent, or convergent, deletion is supposed to be. TBD1, for instance, is not lineage-specific, but it is also not convergent: it occurred once in the common ancestor of lineages 1, 2, and 3, while convergence implies multiple parallel occurrences.

We have changed this and in other places to more evolutionary terms, such as divergent (single event) and convergent (multiple events), or explain exactly what is meant where needed.

l. 118: "where relevant", what does that mean?

This was superfluous to the description and so is now removed.

l. 178ff.: It is not clear to me what issue is addressed by this correction of the pangenome graph. Also here there seems to be some confusion regarding orthologs and paralogs. A gene or IS copy can be present at one locus but absent at another, which is not a mistake of Pangraph that would require correction. It's rather the notion of "truly absent region" which is ambiguous.

We have changed the text to be more specific on the utility of this step. Since it is known that Panaroo mislabels some genes as being absent due to over splitting (see Ceres et al 2022 and our reclassification earlier in the paper), we wanted to see if the same occurred in Pangraph. We have modified the methods text to be more specific (line 181) and in the results included the percentage of total genes/regions affected by this correction.

In relation to copy number, Pangraph is not syntenic in its approach; if a region is present anywhere it is labelled as present in the genome. Pangraph will look for multiple copies of that region (e.g. an IS element) but indeed we did not look for specific syntenic changes across the genomes. This would be a great analysis and something we will consider in the future; we have indicated such in the discussion (line 454).

l. 305: "mislabelled as absent": see above, is this really 'mislabelled'?

See answer to question above

l. 372: "using the approach": something missing here.

This was superfluous to the description and so is now removed.

l. 381: the "additional analysis of paralogous blocks" (l. 381) seems to suffer from the same confusion of ortho- and paralogy described above: no new sub-lineage-specific accessory regions are found presumably because the analysis did consider any copy rather than orthologous copies.

Paralogous copies were looked for by Pangraph, and we did not find any sub-lineage where all members had additional copies compared to other sub-lineages. Indeed, single genomes could have these, and shorter timescales could see a lot of such insertions, but we looked at longer-scale (all genomes within a sub-lineage) patterns and did not find these. These limitations are already outlined in the discussion.

l. 415: see above. There is no diagnosis of a problem that would motivate a "correction". That's different from the correction of the Panaroo results, where fragmented annotations have been shown to be a problem.

Of interest, the refining of regions did re-label multiple regions as being core when Pangraph labelled it as absent from some genomes was at about the same rate as the correction to Pangraph (2% of genes/regions). This indicates there is a stringency issue with pangraph where blocks are mislabelled as absent. The underlying reason or this is not clear but the correction is evidently required in this version of Pangraph.

l. 430ff.: The issue of paralogy and that the "same" gene or region is defined in terms of homology rather than orthology should be addressed here. For me the given evidence does not support the claim that deletion is driving molecular evolution in the MTBC.

As outlined above, indeed paralogy may be driving some elements of the overall evolutionary patterns; our analysis just did not find this. Panaroo without merged paralogs did not find paralogous genes as a main differentiating factor for any sub-lineage. Pangraph also did not find multiple copies of blocks present in all genomes in a sub-lineage. As outlined above, indeed single genomes show such patterns but we did not include single genome analyses here, and outline that as a next steps in the discussion. We have also linked to a recent pangenome paper that showed duplication is present in the pangenome of Mtbc, although not related to any specific lineage (Discussion line 485).

l. 443 ff: "lineage-independent deletions (convergent evolution)": see above, I still think this terminology is unclear

This has now been made clearer to be specifically about convergent and divergent evolutionary patterns.